# Restraint Use for Child Occupants in Dubai, United Arab Emirates

**DOI:** 10.3390/ijerph19105966

**Published:** 2022-05-13

**Authors:** Inam Ahmad, Brian N. Fildes, David B. Logan, Sjaan Koppel

**Affiliations:** Monash University Accident Research Centre, Monash University, Melbourne, VIC 3800, Australia; brian.fildes@monash.edu (B.N.F.); david.logan@monash.edu (D.B.L.); sjaan.koppel@monash.edu (S.K.)

**Keywords:** child restraint systems (CRS), booster seats, child occupant, parents/carers, restraint legislation

## Abstract

The overall objective of the current study was to investigate the behaviours and knowledge of parents/carers in relation to safe child occupant travel in the Emirate of Dubai in the United Arab Emirates (UAE). A community survey was completed by 786 participants who were responsible for the safety of 1614 children (aged 10 years and younger). The survey included questions related to the type, frequency and appropriateness of restraint use for their eldest child. Overall, 24 percent of participants reported that they ‘never/almost never’ restrained their eldest child while travelling in a motor vehicle, with this proportion increasing with child age. For example, though 89 percent of participants reported that they restrained their infants (<1 year) in an ‘appropriate’ restraint for their age, this rate was much lower for children aged between 5 and 7 years (10%). Overall, these findings suggest that a large proportion of child occupants, especially those aged five years and older, are not appropriately restrained in vehicles, and therefore are at an increased risk of death or serious injury in the event of a crash. Future research will validate this self-reported child restraint use data with objective data from observations of real-world child restraint use behaviour in the UAE.

## 1. Introduction

Despite recent advances in vehicle and child restraint system (CRS) design, motor vehicle crashes remain the leading cause of death and serious injuries for those aged between five and 18 years [1]. This is a significant public health problem in most industrialised countries throughout the world, including in the United Arab Emirates (UAE), where the Department of Health of Abu Dhabi—the capital city of the UAE—announced that two out of every three fatally injured children in Abu Dhabi are the result of a motor vehicle crash, around three times the global average [2].

The World Health Organization (WHO) noted that infants and children need a restraint that accommodates their size and weight and can adapt to cope with the different stages of their development [3]. According to the National Highway Traffic Safety Administration (NHTSA), misuse of CRS and booster seats is defined as a characteristic of restraining the child in a CRS/booster seat, which may reduce the protection of the car seat/booster seat in the event of a crash [4]. There are different types of restraints for child occupants, depending on their size and age, including rearward-facing CRS, forward-facing CRS, booster seats and adult seatbelts [1]. The results showed that CRS and booster seats showed effectiveness in reducing nonfatal injury risk and when not seriously misused, CRS and booster seats were associated with a 28 percent reduction in risk for death in children aged 2 through 6 years and when including cases of serious misuse, the effectiveness estimate was slightly lower (21%) [5]. However, the legislation regarding restraint requirements for child occupants when travelling in a motor vehicle differs across the world [1].

In an attempt to prevent motor vehicle crashes and reduce their impact, different UAE government entities have implemented various initiatives and awareness campaigns [6]. One of these initiatives relates to increasing CRS and booster seat use, because they are critical for providing specialised protection to child occupants in the event of a motor vehicle crash [7,8]. Specifically, the UAE’s Federal Traffic Law No. (178) was amended in July 2017 and mandated the use of CRS or booster seats for children aged four years and younger, and stated that child occupants need to be 10 years or older (or at least 145 cm) to travel in the front passenger seat of a motor vehicle [6,9].

Despite motor vehicle crashes being one of the leading causes of death and serious injury for children in the UAE [2], limited research has been conducted in this area [10,11]. Inappropriately restrained child occupants were at nearly twice the risk of injury compared with appropriately restrained children [12]. Grivna et al. [13] reported on child traffic injuries in the UAE and noted that, during 2000–2006, there were 460 traffic deaths among child occupants aged up to 14 years. In addition, the authors noted that the annual average proportion of infants and toddlers (4 years and younger) who died in traffic crashes was 66 percent, which is a high proportion of very young children as traffic victims compared with international data from the World Health Organization [14,15]. According to Grivna et al. [13], from the regional trauma registry of Al Ain city (in the UAE), the traffic injuries accounted 42 percent of those aged 0–19 years hospitalised for injury; of these, 59 percent were vehicle occupants. Most importantly, no child occupants (aged 0–14 years) who died in a crash were found to have used safety restraints and current legislation on safety restraints provides no protection for children and is in any case poorly enforced. Similarly, the findings by Assiry et al. [16], who retrospectively analysed the medical records of paediatric patients aged 16 years and younger admitted to Asir Central Hospital-Abha (main referral unit for southwestern Saudi Arabia) between 2012 and 2014 with traumatic head injuries, reported that motor vehicle-related crashes are the leading cause of all traumatic head injury cases in the region (63%), and 41 percent of these injuries occurred while the child was occupant in the vehicle (21% for those aged between 3 and 7 years and 62% for those aged between 7 and 16 years), and more than half (55%) of all cases were aged between seven and 16 years. A survey conducted by Al-Zahrani et al. [17] found that the leading causes of child injury include motor vehicle crashes in in Riyadh, Saudi Arabia.

One of the earliest studies was conducted in 2008 by Barss et al. [18], who assessed seatbelt wearing rates in Al Ain City (in the UAE) five years after the implementation of seatbelt legislation in the UAE. The data were collected across 2003–2004 at five petrol stations by interviewing the drivers of 500 vehicles (94% male, 47% UAE citizens and 26% off-duty police or military). The study was conducted using an interviewer-administered questionnaire on drivers’ perceptions about seatbelts and reasons for not using them, as well as observing the use of seatbelts by drivers and passengers. The authors noted that seatbelts were only used by 29 percent of drivers. In addition, within the observed vehicles, 382 vehicles also had child occupants (*n* = 876 children) in which CRS and booster seat use was extremely low; only one percent of the child occupants in the rear seat were restrained. Furthermore, 23 percent of the child occupants were travelling in the front passenger seat, almost entirely unrestrained.

In the same year, Grivna et al. [19] conducted a survey to explore the knowledge and attitudes of 260 traffic police officers in Al Ain City regarding CRS and booster seat use (56% UAE citizens). The authors reported that the police officers had misconceptions and limited knowledge about age-appropriate CRS and booster seat use for different child occupant age groups. The authors also reported that, though most participants felt that CRS are necessary for front- and rear-seated child occupants (87%), the driver’s lap was chosen as the safest place in the vehicle for children aged 12 years and younger by three percent of the interviewed officers.

Another survey by Hassan et al. [20] was conducted with road-users (*n* = 2397) throughout all the Emirates to explore their perceptions on different traffic safety items. Of those participants who reported having children, only 39 percent reported that they ‘regularly’ used CRS or booster seats for their children, and 35 percent reported that they ‘never’ used them. In terms of participants’ self-assessment of their awareness level of the traffic laws and regulations, 35, 32 and 24 percent indicated that they had a ‘high’, ‘above average’ and ‘average’ level of awareness, respectively. However, approximately 11 percent of the participants reported that they had ‘no’ level of awareness. Interestingly, 60 percent of participants suggested that campaigning for traffic safety is the most important area that needs improvement, and 53 percent suggested that stricter traffic laws and better enforcement was the second most important area.

Bromfield and Mahmoud [21] used an online survey to explore parents’ use of restraints for their children aged 13 years and younger in Al Ain City from 2013–2014. Based on the responses from 366 parents, the authors reported that less than 20 percent of parents reported that they ‘always’ used a CRS for their child occupant (aged 0–23 months), but that this rate declined depending on whether it was their first-born or subsequent child. The authors also noted that CRS non-use increased as the child’s age increased. For example, for child occupants aged between 2–4 and 4–8 years, the proportion of parents who reported that they ‘never’ used a CRS or booster seat ranged from 37–61 and 53–67 percent, respectively. Additionally, the authors noted that many of the parents who reported that they used a CRS also reported examples of incorrect or inappropriate use. The authors concluded that there is an urgent need for further studies on the facilitators and barriers that are associated with correct and appropriate CRS, booster seat and seatbelt use in the UAE, as well as intervention research to inform locally relevant and culturally appropriate public information campaigns.

Bendak and Alkhaledi [22] used a questionnaire to estimate the rate of CRS and booster seat use among children aged between birth and five years in the UAE. From 2014–2015, the authors interviewed 494 parents (44% male) who were randomly approached out the front of schools, childcare centres and in car parks of shopping centres. Nine percent of the parents reported that they ‘did not’ or ‘rarely’ restrained their children while travelling in a motor vehicle. However, the authors did note that there were differences in CRS and booster seat use across ethnic groups living in UAE. For example, parents with European/North American heritage reported the highest CRS and booster seat use rates (91% reported that they ‘always’ restrained their children) compared to parents with Middle Eastern (68%) or Asian heritage (50%).

Elhalik, El-Atawi and Mahfouz [23] conducted a survey regarding CRS use at the Latifa Women and Children Hospital in Dubai, which is one of the largest maternity and children’s hospitals in the UAE. Over a six-month period in 2017–2018, 201 mothers of newborns from two postnatal wards were recruited. The authors reported that all participants (100%) believed that a CRS is the safest way to transport their child and would like to use it in future. However, the authors noted that 63 percent of the mothers identified reasons for not having a CRS when they left the hospital. For example, 35 percent specified that there is no law enforcing the use of a CRS in Dubai. Interestingly, 75 percent of the mothers supported the need for a law enforcing the use of a CRS or booster seat for child occupants aged four years and younger. The authors noted that, even though all the mothers felt that a CRS was the safest means of transporting their newborn and that they intended to use one in the future, they were not using one. The authors concluded that these findings identified a wide gap between the beliefs and behaviours of mothers in the UAE regarding CRS importance and use, as well as the importance of legislation.

Most recently, Najah, Abuzwidah and Khalil [24] conducted a survey to explore the factors that influence the commitment to wearing a seatbelt in the UAE. The survey was completed by 500 participants (72% female, with 29% having more than three children). The authors reported that 32 percent of parents do not tell their children in the rear seat to wear their seatbelts. The authors also explored whether parents’ behaviours had changed over time. In 2017, for instance, 70 percent of parents reported that they ‘always’ encouraged or taught their children to fasten their seatbelts, whereas two percent reported that they ‘never’ did so. However, by 2018, 75 percent of parents reported that they ‘always’ encouraged their children to use it, whereas one percent reported that they ‘never’ did so. In 2019, 20 percent of parents said that they ‘always’ made sure their children are restrained, but, unfortunately, 32 percent admitted that they ‘never’ did so. In addition, in 2019, there was an increase in the number of participants who reported that they did not use a seatbelt in the rear seat. The authors concluded that following the introduction of Federal Traffic Law No. (178) related to correct and appropriate use of CRS, booster seats and seatbelts, seatbelt-wearing rates increased; however, due to the lack of enforcement by the law, the rates decreased again in 2019.

In her thesis, Abdulrahman [25] investigated the challenges and opportunities for child occupant safety in Abu Dhabi. The study further examined the participants’ perspectives on factors that placed child occupants at risk in society as they presented information about risk factors that they had seen in Abu Dhabi. The majority of the participants (*n* = 7) stated that vehicles and roads presented the highest risk for child occupants. Participants indicated that they dealt with cases of child safety when parents or other adults failed to use child restraints. In some instances, parents reported that they had witnessed instances in which parents would allow a child to use the front passenger seat or sit on the lap of the driver, whether the vehicle was stationary or in motion.

A recent review conducted by Abdullah, Mourad and Muhammad [10] identified all the child occupant safety studies that had been conducted in the UAE during the previous three decades. The results of the review concluded that very few published studies have specifically investigated CRS and booster seat use in the UAE. In addition, the authors noted that, although several promotional campaigns on child occupant safety have been launched in the UAE over the years, there has been limited success in enhancing child occupant safety. The authors hypothesised that one of the explanations for this limited success is that most of the child occupant safety campaigns within the UAE have been replicas of campaigns that had been previously used in other western or developed countries. For example, the campaigns were mostly focused on raising awareness levels restraint use without considering the societal norms and behaviours regarding CRS, booster seat and seatbelt use, misuse and non-use in the UAE.

In addition, Awadalla and Albuquerque [26] analysed data collected from all reported crashes occurring between 2012 and 2017 and reported that many people are still not correctly restraining their children in the Emirate of Abu Dhabi, and this finding may also be relevant to other GCC countries (i.e., Saudi Arabia, Qatar, Kuwait, Oman and Bahrain) due to their similarities in road design, vehicle fleet and driving culture. For example, Alomani et al. [27] assessed the epidemiology, patterns and outcome of trauma in the paediatric population in the Qassim region in Saudi Arabia. The authors investigated 133 cases of children who were admitted to the paediatric intensive care unit (PICU) and paediatric surgery ward and found that these injuries were most frequently the result of road traffic crashes.

### Study Objectives

The overall objective of the current study was to investigate the behaviours and knowledge of parents/carers in relation to safe child occupant travel in the Emirate of Dubai in the United Arab Emirates (UAE). In addition, the current study aimed to explore the following research questions:What is the type and frequency of restraint use (i.e., CRS, booster seat and seatbelt) for child occupants aged 10 years and younger while travelling in their motor vehicles?What are the factors that influence the frequency of restraint use for child occupants aged 10 years and younger?What are the factors that influence appropriate restraint use for child occupants aged 10 years and younger?What are the reasons for restraint non-use?What is the level of awareness regarding the introduction of CRS and boosters seat legislation in the UAE?

## 2. Materials and Methods

### 2.1. Participants

Participants were recruited in this current study if they were: (a) aged 18 years or older, (b) parents/carers of at least one child aged between birth and ten years and (c) resided in the UAE.

### 2.2. Materials

The parents/carers completed a face-to-face community survey (see Appendix A), which contained 23 questions that took approximately 20 min regarding parents’ and carers’ behaviours and knowledge related to safe child occupant travel in the UAE.

#### 2.2.1. Socio-Demographic Characteristics

The parents/carers provided information about their: age, gender, the highest level of completed education, Emirates of residency and adult seatbelt use.

#### 2.2.2. Child Characteristics and Restraint Use

Parents/carers were asked to provide information about the number (and the age and gender) of any children they travel with in their motor vehicle and the type of restraint that the child used most often (i.e., capsule, CRS, booster seat, seatbelt or no restraint). The appropriateness of the restraint type for the eldest child’s age was classified according to the recommendations by NHTSA [4]. According to Goodwin et al. [28], both the American Academy of Paediatrics and NHTSA recommend child occupants stay rear facing for as long as possible until they outgrow the height or weight limits of the restraint, and then use a forward-facing CRS for as long as possible. Another study by Decina et al. [29] stressed the need for back seat positioning for all child ages because it offers independent and additive safety protections in a crash and should be recommended concurrently with each restraint configuration.

#### 2.2.3. Non-Use of Restraint for Children

Participants who indicated that they had not restrained their eldest child were asked to indicate reasons for their non-use.

#### 2.2.4. General Knowledge

Parents/carers were asked about their knowledge regarding the importance of using a restraint appropriate for their child’s age/size, issues regarding CRS or booster seat selection and installation within their vehicle, and knowledge regarding the upcoming introduction of CRS and booster seat legislation in the UAE.

#### 2.2.5. Responses

In responding to these questions, participants were required to limit their responses to just five categories, namely, (1) Always, (2) Almost always, (3) Sometimes, (4) Never and (5) Almost never. In a few instances, they were also asked to elaborate further.

### 2.3. Procedure

The study was approved by the Monash University Human Research Ethics Committee (MUHREC), Melbourne, Australia and was conducted with the assistance of the Roads and Transport Authority (RTA) in Dubai, United Arab Emirates.

A pilot study of the survey was conducted with a focus group of eight participants (mothers of children below 10 years old), and their feedback led to several refinements to improve the clarity of the survey for the targeted audience in the UAE. The final version of the community survey format is provided in the Appendix A.

The face-to-face community surveys were conducted in 2017 by trained interviewers (male and female) in different public places in Dubai (UAE), including: shopping malls, a maternal hospital and several RTA customer service centres. Approval to conduct the community surveys at all locations was granted prior to data collection.

Potential participants (both male and female) were randomly approached at these locations and invited to participate in the community survey, and an informed consent was obtained from all participants involved in the study. Each survey took approximately 20 minutes to be completed and was administered in either Arabic or English, depending on the participant’s preference.

It is important to note that this study was conducted prior to the introduction of the UAE’s amended Federal Traffic Law No. (178) in July 2017 that mandated the use of CRS or booster seats for child occupants aged four years and younger [9].

### 2.4. Data Analysis

Descriptive statistics were conducted to describe the sample. A series of chi-square analyses were conducted to explore the parents’/carers’ behaviours and knowledge relating to safe child occupant travel in the Emirate of Dubai in the UAE. All statistical analyses were conducted using IBM SPSS Statistics for Windows, version 28 (IBM Corp., Armonk, NY, USA).

## 3. Results

The findings for this study are presented in five main sections: (1) Participants’ socio-demographic characteristics and their seatbelt use; (2) Characteristics of the participants’ eldest child and the type and frequency of their restraint use while travelling in a motor vehicle, (3) Factors that influence the frequency and appropriate restraint use for child occupants while travelling in a motor vehicle; (4) Reasons for non-use of restraints, and (5) Participants’ level of awareness of the introduction of CRS and booster seat legislation in the UAE.

### 3.1. Participants’ Socio-Demographic Characteristics

The survey was completed by 786 parents/carers who were responsible for 1614 children (aged between birth and 10 years). Table 1 shows that most participants: were aged between 26 and 35 years (48.5%); were female (58.3%); had completed an undergraduate university degree (67.9%); were not UAE nationals (61.1%); ‘always’ used their seatbelt while travelling in a motor vehicle (72.1%) and lived in the Emirate of Dubai (79.0%).

### 3.2. Child Characteristics and Restraint Use

Participants had a mean of just over two children within their families. Table 2 shows that most of the participants’ eldest children were: aged between one and seven years (65.6%); male (55.7%); restrained by a seatbelt (48.0%) and ‘always’ restrained while travelling in a vehicle (44.4%).

### 3.3. Factors That Influence Frequency of Restraint Use for Child Occupants

This study investigated the relationships between participants’ characteristics and the frequency of the restraint use for their eldest child while travelling in a motor vehicle.

#### 3.3.1. Participants’ Age Group by Frequency of Child’s Restraint Use

There was a significant relationship between participants’ age group and the frequency of restraint use for their eldest child (χ^2^(9) = 38.9, *p* < 0.001) (See Figure 1). Younger participants (i.e., aged 18–25 years) were more likely to report that they ‘always’ restrained their eldest child while travelling in a motor vehicle, whereas older participants (i.e., aged 36–49 and 50+ years) were more likely to report that they ‘never/almost never’ restrained their eldest child while travelling in a motor vehicle.

#### 3.3.2. Participants’ Gender by Frequency of Child’s Restraint Use

There was also a significant relationship between participants’ frequency of restraint use for the eldest child and the participant’s gender (χ^2^(3) = 13.0, *p* < 0.001) (see Table 3). Female participants were more likely to report that they ‘always’ restrained their eldest child while travelling in a motor vehicle (48.7%) compared with male participants (38.4%).

#### 3.3.3. Participants’ Education Level by Frequency of Child’s Restraint Use

There was also a significant relationship between participants’ frequency of restraint use for the eldest child and their education level (χ^2^(6) = 46.3, *p* < 0.001) (see Table 4). Participants who had completed a postgraduate or undergraduate degree were more likely to report that they ‘always’ restrained their eldest child while travelled in a motor vehicle (50.0%, 47.0% respectively) compared to participants who had completed middle/high school (34.4%).

#### 3.3.4. Participants’ Nationality by Frequency of Child’s Restraint Use

There was a significant relationship between participants’ frequency of restraint use for their eldest child and their nationality (χ^2^(3) = 26.9, *p* < 0.001) (see Table 5). Participants with UAE nationality were less likely to report that they ‘always’ restrained their eldest child (28.9%) compared with participants from other nationalities (71.1%).

#### 3.3.5. Participants’ Seatbelt Use by Frequency of Child’s Restraint Use

There was a significant relationship between participants’ frequency of restraint use for the eldest child and the frequency of their own seatbelt use (χ^2^(9) = 69.4, *p* < 0.001) (see Table 6). Participants who reported that they ‘always’ restrained their eldest child were most likely to report that they ‘always’ used their own seatbelt (51.1%) and least likely to report that they ‘never/almost never’ used their seatbelt (18.2%).

#### 3.3.6. Participants’ Emirate of Residency by Frequency of Child’s Restraint Use

There was no significant relationship between participants’ frequency of restraint use for the eldest child and their Emirate of residency (χ^2^(3) = 3.6, *p* = 0.306) (see Table 7). Participants who were residents in Dubai were just as likely to report that they ‘always’ restrained their eldest child (43.8%) as participants who were residents of the ‘Other emirates’ (44.4%).

### 3.4. Child-Related Factors That Influence Restraint Use

#### 3.4.1. Child’s Age Group by Frequency of Restraint Use

There was a significant relationship between the eldest child’s age group and the frequency with which they were restrained, χ^2^(9) = 110.7, *p* < 0.001 (see Figure 2). Participants were more likely to report that they ‘always’ restrained their eldest child if they were in a younger age group (<1 year: 75.0%, 1–4 years: 61.6%) than if they were in an older age group (5–7 years: 39.5%; 8–10 years: 26.1%).

#### 3.4.2. Child’s Age Group by Type of Restraint Use

There was a significant relationship between the eldest child’s age group and their restraint type, χ^2^(12) = 592.6, *p* < 0.001 (see Figure 3). Participants were more likely to report that they had restrained their eldest child in a capsule for those aged one year or younger (77.8%), in a CRS if aged between one and four years (53.8%) and in a seatbelt for those aged five years and older. It was also interesting to note that the likelihood of not being restrained increased with the eldest child’s age.

#### 3.4.3. Childs’ Age Group by Appropriateness of Restraint Type

There was a significant relationship between the eldest child’s age group and the appropriateness of their restraint type, χ^2^(3) = 231.0, *p* < 0.001 (see Figure 4). Participants were more likely to report that they had restrained their eldest child in an ‘appropriate’ restraint (based on their age) if their eldest child was aged less than one year (88.9%) and least likely if their child was aged between five and seven years (9.9%).

### 3.5. Non-Use of Restraint for Children

One hundred and eighty-seven participants (i.e., 23.8% of the total sample) reported that they ‘never/almost never’ restrained their eldest child while travelling in a motor vehicle. These participants were asked if there was a specific reason for ‘never/almost never’ restraining their eldest child. The most common response was ‘child’s discomfort’ (46.5%) (see Figure 5). Other reasons included ‘lack of CRS importance’ (17.6%), ‘insufficient space’ in the vehicle (11.2%), ‘nearby destinations’ (3.7%), ‘no legal penalties’ (2.1%) and ‘CRS are too expensive’ (1.6%).

### 3.6. Parents’ Awareness of the Introduction of CRS and Boosters Seat Legislation

Participants were asked if they were aware of the CRS and booster seat legislation that was about to be introduced in the UAE. More than two thirds of participants (77.6%) reported that they were aware that there was legislation which was about to be introduced that required the driver to restrain their child(ren) while travelling in a moving vehicle. There was a significant relationship between parents’ knowledge of CRS and booster seat legislation that was about to be introduced and the frequency of restraint use for their eldest child, χ^2^(3) = 35.3, *p* < 0.001 (see Table 8). Participants who reported that they were aware of this legislation were less likely to report that they ‘never/almost never’ restrained their eldest child (19.0%) compared to participants who were not aware of this legislation (40.3%).

## 4. Discussion

The overall objective of the current study was to investigate the behaviours and knowledge of parents/carers in relation to safe child occupant travel in the Emirate of Dubai in the United Arab Emirates (UAE). Specifically, this study implemented a community survey to study participants’ responses to the five research questions set up to address this objective:What is the type and frequency of restraint use (i.e., CRS, booster seat and seatbelt) for child occupants aged 10 years and younger while travelling in their vehicles?What are the factors that influence the frequency of restraint use for child occupants aged 10 years and younger?What are the factors that influence appropriate restraint use for child occupants aged 10 years and younger?What are the reasons for restraint non-use?What is the level of awareness regarding the introduction of CRS and booster seat legislation in the UAE?

The results from this study will provide insights to improve child occupant safety in the UAE, which is important given that two out of every three fatally injured children in Abu Dhabi have been in a motor vehicle crash, around three times the global average [2].

### 4.1. Type and Frequency of Restraint Use

Overall, the study shows that less than half (44.4%) of the participants reported that they ‘always’ restrained their eldest child while travelling in a motor vehicle. In addition, 23.8 percent reported that they ‘never/almost never’ restrained them. These findings are consistent with the findings from other, similar studies in the Arabian Gulf countries [15,17,24,30,31,32,33].

Not surprisingly, as shown in Figure 3, there was a significant relationship between the eldest child’s age group and their restraint type. Children were most likely to be restrained in a capsule if aged one year or younger, in a CRS if aged between one and four years, and in a seatbelt if aged five years and older. This study finding is of great concern since, as mentioned earlier, according to the World Health Organization (WHO) reports, road traffic injury is the primary cause of death for children five years and older [1]. It is interesting to note that the likelihood of not being restrained increased with the increase of child age from 5 years in this study, a practice that puts these children at a high risk of fatal injury in the event of a crash.

### 4.2. Significant Factors That Influence Frequency of Restraint Use for Child Occupants

The current study also identified some of the participants’ socio-demographic factors that were significantly associated with the frequency of restraint use for their eldest child. The first factor was the participant’s age; as shown in Figure 1, younger participants (i.e., aged 18–25 years) were more likely to report that they ‘always’ restrained their eldest child compared to older participants (i.e., aged 36–49 and 50+ years). These findings are consistent with the outcomes that were reported by Zaidan [30], in which younger parents were more likely to report that they restrained their children compared to older parents. However, it is inconsistent with the findings of Bendak and Alkhaledi [22] and Fildes et al. [32], who reported that older participants were more likely to restraint their children. The second factor was participant gender, as shown in Table 3, with female participants more likely to report that they ‘always’ restrained their eldest child while travelling in a motor vehicle compared with male participants. This finding is consistent with Bendak and Alkhaledi [22] and with Liu et al. [34], in which more female parents answered ‘yes’ to CRS use for their children (<12 years) compared to male parents. It is also consistent with the findings of Lei et al. [35], in which the use of CRS was statistically significantly higher among female participants. Awadalla and Albuquerque [26] found that Emirati male drivers had a higher risk of being involved in a fatal road crash in Abu Dhabi, potentially explaining the results of this study in relation to gender. Our findings also show the third factor is related to participant education, which is similar to the findings of Bendak and Alkhaled [22] and Fildes et al. [32], in which participants holding a higher educational degree were more likely to report that they ‘always’ restrained their children. We also found that participants’ culture and nationality might play a role, as a fourth factor, with participants with UAE citizenship less likely to report that they ‘always’ restrained their eldest child compared with participants from other nationalities. The last factor was related to a possible correlation between parental seatbelt use and the use of CRS for their child, which is consistent with the findings of Fildes et al. [32].

The current study also identified some child characteristics that were significantly associated with the frequency of restraint use. Participants were more likely to ‘always’ use a restraint for younger children (i.e., aged less than one year), and restraint use decreased with the child’s age. Similarly, Fildes et al. [32] observed that the use of restraints for older children was generally less than those of younger children in rear seats in Saudi Arabia, putting children aged five years and above at the highest risk of death or serious injury in the event of a crash.

### 4.3. Factors That Influence Appropriate Restraint Use for Child Occupants

In this study, the results showed that there was a significant relationship between the eldest child’s age group and the appropriateness of their restraint. As shown earlier in Figure 4, the proportion of appropriately restrained children was highest for those aged less than one year and lowest for those aged between five and seven years. Although almost ninety percent of the interviewed participants reported that they appropriately restrained their infants, this percentage decreased dramatically with five- to seven-year-old children, with only circa ten percent of the participants claiming to use an appropriate restraint. A previous study in the UAE by Elhalik et al. [23] found that only 45.7 percent of parents claimed that they used CRS when leaving the hospital with their newborn child.

### 4.4. Non-Restraint Use Factors

The reasons for not using the restraint for children were reported by the parents/carers and outlined in Figure 5 above. Almost half of the participants (46.5%) reported that they did not restrain their children due to the discomfort of their child, and 17.6 percent of participants attributed the non-use of restraint to the lack of importance of CRS. These findings are consistent with the reasons given by participants in the study by Fildes et al. [32], who believed that children were adequately restrained by their parents (i.e., in their arms or laps), and they were well-protected ‘spiritually’. A considerable portion (31.3%) in Fildes et al. [32] study opined that holding the newborn in their arms was safer; about 18 percent considered seatbelts unnecessary for children, and more than half (54.2%) did not have a child safety seat fitted to their vehicle while leaving the hospital with their newborn. On the other hand, our findings regarding the discomfort of the child and the use of restraint is inconsistent with Fong et al. [36], who found that there was no significant relationship between either parent-reported comfort and restraint misuse or parent-reported comfort and age-appropriate restraint choice. ‘Insufficient space’ in the vehicle (family size, number of children in the family, vehicle size) was the third highest (11.2%) reason given for non-restraint use in our study, which is consistent with Zaidan [30] and Nelson et al. [37], who linked low use of CRS with both family and car size.

### 4.5. Knowledge of the Impending CRS and Boosters Seat Legislation in the UAE

At the time of this study, the new CRS and booster seat legislation had not yet been implemented in the UAE. However, the results showed that the majority of participants (77.6%) were aware that the legislation was about to be introduced, and those who were aware of the impending law were more likely to report that they ‘always’ restrained their eldest child compared to participants who were not aware. A previous study by Fildes et al. [32] recommended to extend the current legislative requirements in the KSA to mandate the use of CRS for children to improve compliance towards CRS use.

## 5. Challenges and Opportunities

The data in the current study were collected through a face-to-face community survey, and the survey team was trained by a research team from Monash University Accident Research Centre (MUARC) in Melbourne, Australia, with the collaboration of the Roads and Transport Authority (RTA) in Dubai, United Arab Emirates. However, it is important to note that the data are self-reported and therefore may be subject to biases. The sample also might not be considered as representative of the UAE population, and it is important to note that the survey responses have not been validated in this study; this is a potential limitation of the current study. The focus of the literature review was on the most relevant studies from UAE and GCC countries (i.e., Saudi Arabia, Qatar, Kuwait, Oman and Bahrain) due to their similarities in road design, vehicle fleet and driving culture. However, the inconsistencies between our findings and previous studies may be due to differences in survey methodology or participant demography. One way to validate some of our findings would be to conduct an observation study to observe real-world restraint use and compare it with the behaviours that were self-reported during the community survey. Finally, the data were collected in 2017, and it will be important to replicate this study after the introduction of the new restraint legislation to determine whether the legislation has improved restraint use for child occupants aged 10 years and younger. We believe that this study is novel because it is the first study that has specifically investigated both the frequency and appropriateness of restraint use for children aged 10 years and younger in the UAE that takes demographic and cultural influences into consideration. 

## 6. Conclusions

The findings of a study entitled “Epidemiology and Prevention of Child Injuries in the United Arab Emirates” were discussed during a policy session for injuries among children at the Arab Children Health Congress held in Dubai in 2010, and, thereafter, the outcomes were published under the title ‘Accidental injuries prime death cause of children in the United Arab Emirates’, in which a policy roundtable recommended a key policy that included strengthening and combining efforts in the areas of surveillance and research, building capacity in these areas, developing a regional strategy on child injury prevention, identifying clear indicators and targets, as well as creating a culture of safety through awareness and education campaigns towards families and children [38].

The current study has identified several important findings that should be the focus of future awareness programs. Of most concern is the low proportion of child occupants reportedly travelling in an appropriate restraint in general, and specifically for children aged between 5 and 7 years. Education and awareness programs that describe the dangers of premature graduation to an adult seatbelt are urgently needed. Any future programs also need to focus on increasing the awareness of specific demographic segments (i.e., older parents, males, UAE nationals, parents with lower levels of education). There is also a need to discuss restraint discomfort and increase awareness regarding appropriate restraint use in addition to exploring options for large families to travel safely with their children in motor vehicles. The outcomes of the current study have been shared with the Roads and Transport Authority (RTA) in Dubai, so they can use the outcomes for child safety programs.

Given the predominant role of social media in the UAE, the use of this medium for the dissemination of appropriate messages should be explored. Future research is currently underway to validate the self-reported data in this study with observations of on-road behaviour. These findings will be used identify recommendations that can help to increase correct and appropriate restraint use for child occupants while travelling in motor vehicles throughout the UAE. In addition, it will be important to replicate this study after the introduction of the new restraint legislation to determine whether the legislation has improved restraint use for child occupants aged 10 years and younger.

## Figures and Tables

**Figure 1 ijerph-19-05966-f001:**
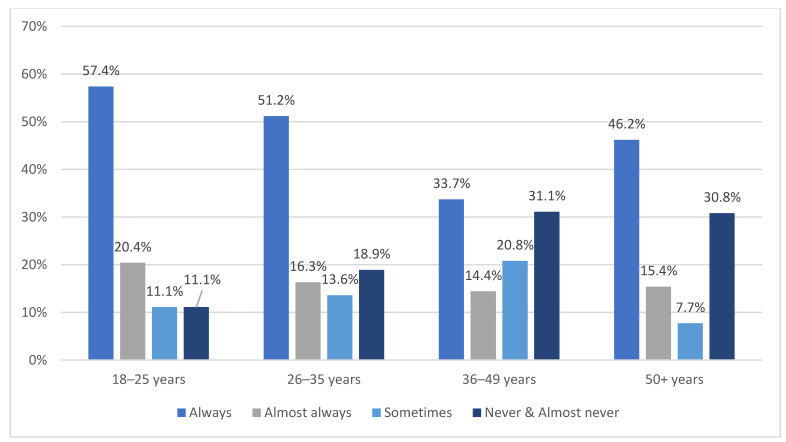
Respondents’ age group and frequency of restraint use for their eldest child while travelling in their vehicle.

**Figure 2 ijerph-19-05966-f002:**
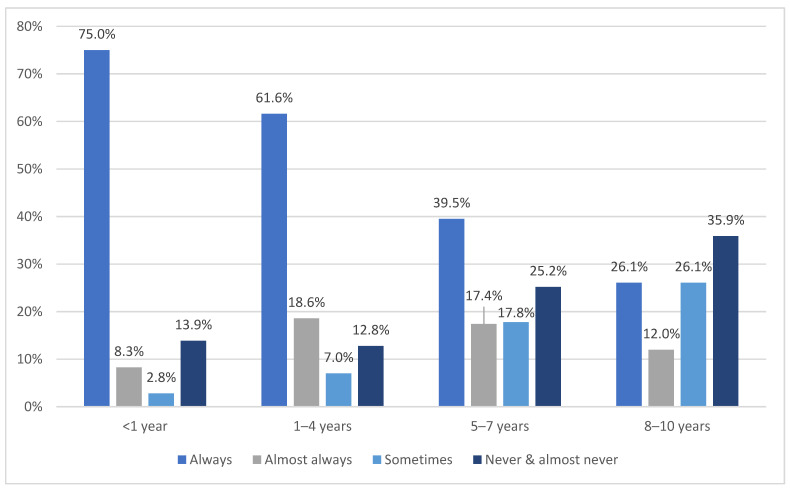
Child’s age group and frequency of restraint use while travelling in a motor vehicle.

**Figure 3 ijerph-19-05966-f003:**
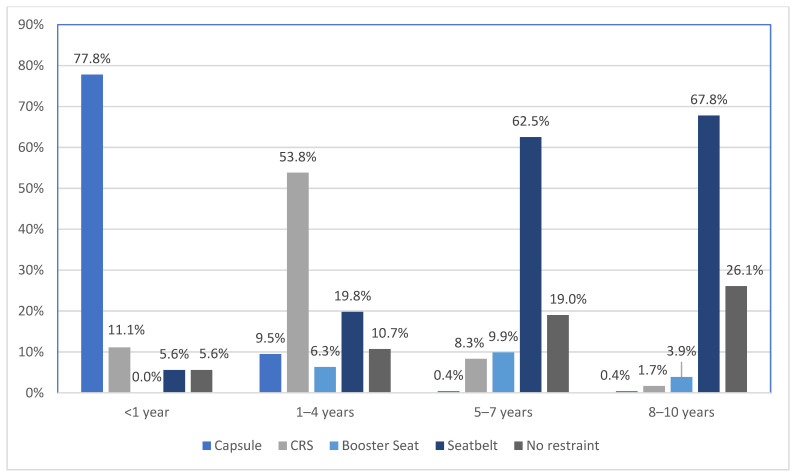
Type of restraint by age group for the eldest child.

**Figure 4 ijerph-19-05966-f004:**
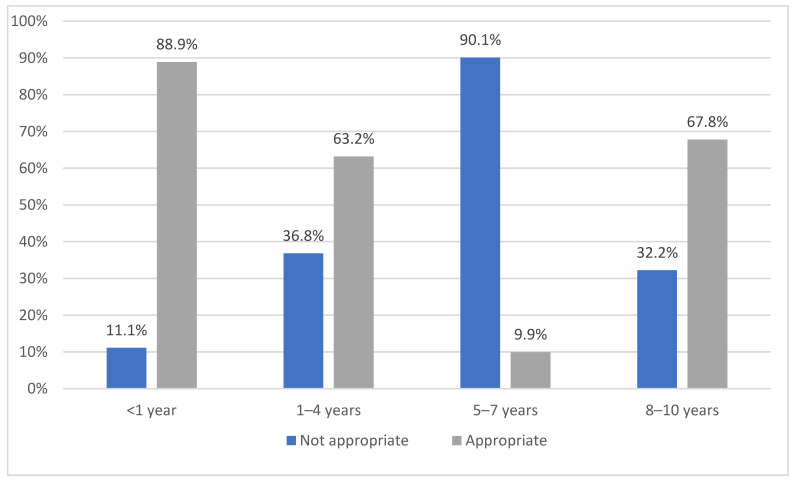
Appropriateness of restraint use by age group for the eldest child.

**Figure 5 ijerph-19-05966-f005:**
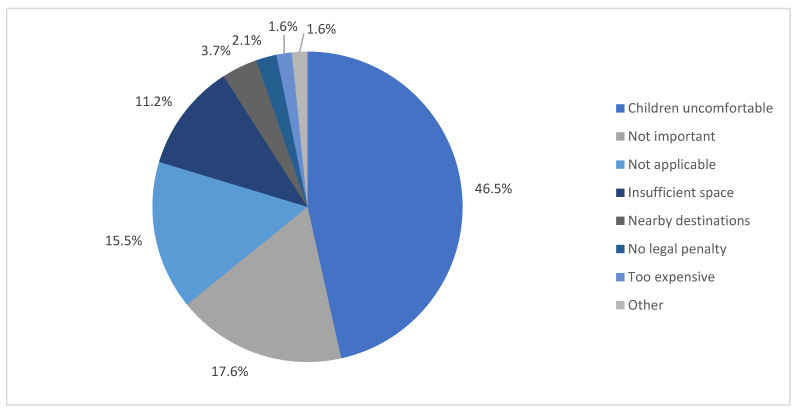
Reasons for ‘never/almost never’ restraining their eldest child while travelling in a motor vehicle (*n* = 187).

**Table 1 ijerph-19-05966-t001:** Participants’ socio-demographics characteristics and restraint use.

Socio-Demographics Characteristics		% (n)
Age (years)	18–25	6.9% (54)
	26–35	48.5% (381)
	36–49	39.7% (312)
	50+	5.0% (39)
Gender	Male	41.7% (328)
	Female	58.3% (458)
Highest level of completed education	Middle/High School	22.9% (180)
	Undergraduate degree	67.9% (534)
	Postgraduate degree	9.2% (72)
Nationality	UAE	38.9% (306)
	Other nationality	61.1% (480)
Adult seatbelt use	Always	72.1% (567)
	Almost always	14.0% (110)
	Sometimes	9.9% (78)
	Never/Almost never	3.8% (30)
Emirate of residency	Dubai	79.0% (621)
	Other Emirate	21.0% (165)

**Table 2 ijerph-19-05966-t002:** Eldest child characteristics and type and frequency of restraint use.

Child Characteristics		% (*n*)
Age group	<1 year	4.6% (36)
	1–4 years	32.8% (258)
	5–7 years	32.8% (258)
	8–10 years	29.8% (234)
Gender	Male	55.7% (438)
	Female	44.3% (348)
Restraint type	Capsule	6.9% (54)
	CRS	21.0% (165)
	Booster seat	6.4% (50)
	Seatbelt	48.0% (377)
	No restraint	17.8% (140)
Restraint frequency use	Always	44.4% (349)
	Almost always	15.8% (124)
	Sometimes	16.0% (126)
	Never/Almost never	23.8% (187)

**Table 3 ijerph-19-05966-t003:** Participants’ gender and frequency of restraint use for their eldest child while travelling in a motor vehicle.

Frequency of Restraint Use	Gender	
	Female % (*n*)	Male % (*n*)
Always	48.7% (223)	38.4% (126)
Almost always	14.6% (67)	17.4% (57)
Sometimes	16.8% (77)	14.9% (49)
Never/Almost never	19.9% (91)	29.3% (96)

**Table 4 ijerph-19-05966-t004:** Participants’ education level and frequency of restraint use for their eldest child.

Frequency of Restraint Use	Education Level		
	Middle/High School % (*n*)	Undergraduate Degree % (*n*)	Postgraduate Degree % (*n*)
Always	34.4% (62)	47.0% (251)	50.0% (36)
Almost always	10.0% (18)	18.0% (96)	13.9% (10)
Sometimes	13.3% (24)	17.2% (92)	13.9% (10)
Never/Almost never	42.2% (76)	17.8% (95)	22.2% (16)

**Table 5 ijerph-19-05966-t005:** Participants’ nationality and frequency of restraint use for their eldest child while travelling in a vehicle.

Frequency of Restraint Use	Nationality	
	UAE % (*n*)	Other Nationality % (*n*)
Always	28.9% (101)	71.1% (248)
Almost always	44.4% (55)	55.6% (69)
Sometimes	49.2% (62)	50.8% (64)
Never/Almost never	47.1% (88)	52.9% (99)

**Table 6 ijerph-19-05966-t006:** Participants’ seatbelt use and frequency of restraint use for their eldest child while travelling in a vehicle.

Frequency of Restraint Use	Participants’ Seatbelt Use			
	Never & Almost Never % (*n*)	Sometimes % (*n*)	Almost Always % (*n*)	Always % (*n*)
Always	25.8% (8)	20.5% (16)	31.8% (35)	51.1% (290)
Almost always	6.5% (2)	11.5% (9)	24.5% (27)	15.2% (86)
Sometimes	16.1% (5)	19.2% (15)	16.4% (18)	15.5% (88)
Never & almost never	51.6% (16)	48.7% (38)	27.3% (30)	18.2% (103)

**Table 7 ijerph-19-05966-t007:** Participants’ Emirate of residency and frequency of restraint use for their eldest child.

Frequency of Restraint Use	Emirate of Residency	
	Dubai % (*n*)	Other Emirates % (*n*)
Always	43.8% (14)	44.4% (335)
Almost always	6.3% (2)	16.2% (122)
Sometimes	25.0% (8)	15.6% (118)
Never/Almost never	25.0% (8)	23.7% (179)

**Table 8 ijerph-19-05966-t008:** Participant’s awareness of the upcoming introduction of CRS and booster seat legislation and frequency of restraint use for their eldest child while travelling in a motor vehicle.

Frequency of Restraint Use	Awareness of Introduction CRS and Booster Seat Legislation	
	Aware (Yes) % (*n*)	Not Aware (No) % (*n*)
Always	47.4% (289)	34.1% (60)
Almost always	17.2% (105)	10.8% (19)
Sometimes	16.4% (100)	14.8% (26)
Never/Almost never	19.0% (116)	40.3% (71)

## Data Availability

The dataset supporting the conclusions of this article is available from the corresponding author on reasonable request.

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
