# Peer review of "Restraint Use for Child Occupants in Dubai, United Arab Emirates"

_ijerph, 2022, doi:10.3390/ijerph19105966_

Round 1
Reviewer 1 Report
Dear Sirs, the work with the modifications made is ready to be published. receive a cordial greeting and congratulations to the authors for their contributionAuthor Response
Thank you very much. We appreciate your comments and believe that the revisions have improved the quality of the manuscript.
Reviewer 2 Report
In the final References, the authors cannot put "et al.", they must put the name of all the authors of the publications. You should put "et al." only throughout the text (and not in the final references), only when the authors of the publication were more than 3 authors. See for example the final references:
- "5. Elliott, M.R., et al., Effectiveness of child safety seat ..."
- "7. Anund, A., et al., Child safety in cars ..."
- "8. Mendoza, A.E., et al., The worldwide approach to Vision Zero ..."
- "12. Durbin, D.R., et al., Effects of seating position and appropriate ..."
- "16. Assiry, K., et al., Traumatic head injuries in children: experience from ..."
- "17. Al-Zahrany, M.S., et al., Self-reported Unintentional Injuries ..."
- "18. Barss, P., et al., Prevalence and issues in non-use of safety belts and child restraints ..."
- "28. Goodwin, A.H., et al., Countermeasures That Work: A Highway ..."
And still others!
Authors should review all references and correct them before being accepted for publication!
Author Response
Thank you for the comment.
We have reviewed all the references and have corrected them.
Reviewer 3 Report
I accept your changes to the article. I still have doubts about the advisability of publishing these results. However, I hope that you will show what changes have taken place in your next publication.
Please reconsider whether very short chapters are necessary: 2.2.1; 2.2.3; 2.2.4; 3.3.2; .....
Author Response
Thank you very much. We appreciate your comments and believe that the revisions have improved the quality of the manuscript. We are also very keen to validate some of these findings via a real-world observation study. We are also keen to replicate this study following the introduction of the new restraint legislation to determine whether restraint use for child occupants aged 10 years and younger has improved.
Regarding the short chapters, appreciate this comment, however we are following the Author guidelines for the Journal and therefore must include the sub-headings.
This manuscript is a resubmission of an earlier submission. The following is a list of the peer review reports and author responses from that submission.
Round 1
Reviewer 1 Report
In my opinion, the article needs to improve the following aspects:
Rewrite the keywords to remove the compound ones and reduce them, Without a doubt reduce the abstract, it is too long, state the main conclusion of the work in the abstract. Define the main objective of the work in the introduction clearly and not Quintuple objective and establish some secondary objectives, as well as referencing and expanding the international bibliography. And include international bibliography.
BROAD AND SPECIFIC COMMENTS.
In my opinion, you have to reduce keywords, as well as reduce their size, certainly reduce the summary, it is too long.
In the introduction it is not clearly stated what the main objective of the work is, it is intuited that it is: The "general objective of the present study was to carry out a community survey to investigate", but it is not stated clearly, just as this does not may be the objective (conduct a survey), this should be modified, and put as they say in the summary: "investigate the behaviors and knowledge of parents in relation to the safe child travel of the occupants in the Emirate of Dubai in the United Arab Emirates United (UAE)”.
Define the main objective of the work in the introduction clearly and not Fivefold objective and establish some secondary objectives, as well as referencing and expanding the international bibliography. And include international bibliography.
No research hypothesis is clearly stated, which means that the results remain up in the air, and therefore they cannot be taken for granted, they must state and list the research hypotheses, detailing them since they do not appear in the work.
Reviewer 2 Report
The authors present an excellent Abstract in the article. It is very complete and has all the necessary information for the scientific community, regarding the research developed by the authors.
[line 92] The authors refer to "… Grivna and colleagues [16] …" in the article, which is not correct. It would be more correct if authors wrote in the article: "… Grivna et al. …"
[line 111] The authors refer to "… Bromfield and colleagues [18] …" in the article, which is not correct. It would be more correct if authors wrote in the article: "… Bromfield and Mahmoud [18] …"
[line 125] The authors refer to "… Bendak and colleagues [19] …" in the article, which is not correct. It would be more correct if authors wrote in the article: "… Bendak and Alkhaledi [19] …"
[line 135] The authors refer to "… Elhalik and colleagues [20] …" in the article, which is not correct. It would be more correct if authors wrote in the article: "… Elhalik, El-Atawi, and Mahfouz [20] …"
[line 150] The authors refer to "… Najah and colleagues [21] …" in the article, which is not correct. It would be more correct if authors wrote in the article: "… Najah, Abuzwidah, and Khalil [21] …"
[line 175] The authors refer to "… Abdullah and colleagues [10] …" in the article, which is not correct. It would be more correct if authors wrote in the article: "… Abdullah, Mourad, and Muhammad [10] …"
[line 187] The authors refer to "… Awadalla and colleagues [23] …" in the article, which is not correct. It would be more correct if authors wrote in the article: "… Awadalla and Albuquerque [23] …"
[line 192] The authors refer to "… Alomani and colleagues [24] …" in the article, which is not correct. It would be more correct if authors wrote in the article: "… Alomani et al. [24] …"
[line 197] It would be better if the authors put the " Study Objectives ", inside the Materials and Methods.
What software did the authors use to statistically analyse and process the data collected in the survey?
The authors rightly point out that they initially validated the survey with a pilot group of 10 people. However, it is not indicated how the authors validated the data collected in the survey.
It would be very important that the authors at the end recommend future works, so that the scientific community could continue their investigation. What is the work that the authors would like to see developed by other researchers, so that they could see the consolidation of this research?
In the final References, the authors cannot put "et al.", they must put the name of all the authors of the publications. You should put "et al." only throughout the text (and not in the final references), only when the authors of the publication were more than 3 authors. See for example the final references 24, 25 and 26.
Reviewer 3 Report
The assessed work entitled Restraint Use for Child Occupants in Dubai, United Arab Emirates is concerned with parents' knowledge and behaviour about child travel safety in motor vehicles. This issue is very important and timely. Every year many children die as a result of road traffic accidents. Some of these are the direct result of inadequate child restraint during travel. I believe that every initiative aimed at improving transport safety is very important. Children are very often unaware of the risks. The driver or supervisor therefore takes full responsibility for the safety of travellers.
1. The introduction provides sufficient information. It is presented in a clear way. The only criticism is the very small number of recent publications. Most of the literature is older than 5 years. In recent years, this subject has not been popular in scientific research. Most of the issues are regulated by legislation.
2. The abstract is over 300 words. I propose to shorten it to the required 200 words.
3. The aim of the study was to find out the practices and knowledge of the local community regarding the safety of children travelling in motor vehicles. Only parents/carers of children aged up to 10 years participated in the survey. It is not clear to me what scientific problem the authors want to solve. The literature review gives a broad description of the problem but does not identify gaps in the current state of knowledge. Most of your results confirm what others have already done. Which is the novelty of this work.
4. Why did you wait from 2017 to 2022 to publish the results of the study? Are these results still valid? There were legal changes to this in 2017. Row 262. can show how these have affected child travel safety. How carers have changed their behaviour.
5. Is the sample taking part in the survey representative? How was it assessed?
6. Was an attempt made to assess the reliability of the survey before further analysis of the results? Line 530.
7. Why did only mothers of children participate in the pilot study? Line 250.
8. The methodology lacks a description of the statistical methods used in the survey.
9. The tables use once the description "% (n)" and once "Percentage (n)". Please use one description.
10. The reporting of the average number of children I consider unnecessary. Line 287
11. How the survey results improve safety. Line 422
12. The discussion lacks an attempt to explain the differences between the results obtained and the literature. Why in the United Arab Emirates young parents are more likely to restrain their children and in the work of Bendak and Alkhaledi [19] and Fildes [29] not.
13. What is new about this work and relevant to science?
14. Why are acknowledgements included in the Patents section?
15. The authors' contribution to the work is not presented in the paper.
16. The references in the literature lack number 1. the number 38 is shifted.
The subject matter is very interesting and important. I believe that you have very good material. I only suggest that you revise its presentation.
Reviewer 4 Report
My comments are added directly into the pdf file of the paper, attached.
